# “Geriatric Proximity” Intervention in COVID-19 Context: Contribution to Reducing Loneliness and Improving Affectivity

**DOI:** 10.3390/geriatrics8020039

**Published:** 2023-03-21

**Authors:** Bruno Morgado, Cesar Fonseca, Anabela Afonso, Pedro Amaro, Manuel Lopes, Lara Guedes de Pinho

**Affiliations:** 1Nursing Department, University of Évora, 7000-801 Évora, Portugal; 2Garcia de Orta Hospital, EPE, 2805-267 Almada, Portugal; 3Escola de Doctorat, Universitat Rovira i Virgili, 43005 Tarragona, Spain; 4Comprehensive Health Research Centre, University of Évora, 7000-801 Évora, Portugal; 5Department of Mathematics, School of Sciences and Technology, University of Évora, 7000-671 Évora, Portugal; 6Center for Research in Mathematics and Applications (CIMA), IIFA, 7000-671 Évora, Portugal; 7Institute for Advanced Studies and Research, University of Évora, 7004-516 Évora, Portugal

**Keywords:** affective symptoms, functioning, loneliness, long-term care, healthy aging, nursing models, social support

## Abstract

(1) Background: The pandemic context has limited the social and family contacts of institutionalized older adults, and intervention is urgently needed. The aim of this study is to assess the impact of the implementation of a “Geriatric Proximity” intervention on the functioning, satisfaction with social support, affective experience, and feelings of loneliness of institutionalized older adults in the times of the pandemic. (2) Methods: This is a pilot study. An experimental group (subject to the “Geriatric Proximity” intervention) and a control group were constituted. Four assessment instruments were applied to both groups: the satisfaction with social support scale; the elderly nursing core set; the positive and negative affect schedule; and the UCLA loneliness scale. (3) Results: The control group shows no differences between the three measurement instants, while the experimental group shows between first and third measurements (all *p* < 0.05). We observed a reduction in the scores of loneliness scale, negative affect, and cognition functioning and an increase in satisfaction with social support and positive affect. (4) Conclusions: The intervention “Geriatric Proximity” showed a positive contribution by decreasing loneliness and increasing affectivity, satisfaction with social support, and cognitive function during the pandemic period.

## 1. Introduction

In December 2019, in Wuhan, China, a SARS-CoV-2 virus causing an acute respiratory syndrome, named COVID-19, caused the World Health Organization to declare a pandemic on 11 March 2020, throwing the world population into their homes to control the transmission of SARS-CoV-2 [1]. The most affected population, due to fragility, was undoubtedly the older adults, especially in residential homes [2]. Data from the Center for Disease Control and Prevention in the United States indicate that 17% of the infections were in the population over 65 years age, but 80% of the deaths occurred in this population [3,4]. To protect the institutionalized older adults, the government of Portugal decreed the cancellation of visits in residential homes for older adults.

International studies indicate that a break in the family or social contact between the older adults and the outside world compromises the mental health of them, thus increasing social isolation, loneliness, anxiety, and depression and reducing cognitive ability [5,6]. The pandemic has worsened the mental health of older people and increased depressive symptoms [7]. Nurses should be aware that visit restrictions can negatively affect patients and families and should adapt care delivery to compensate for such effects [6]. To respond to negative effects related to the deprivation of visits and contact with the outside world, it is important to follow the principles of active and healthy aging, highlighting the importance of affection and social support for the maintenance of mental health in older adults [8]. Social support and interactions are fundamental for quality of life in older adults, considering that they improve functional ability as a predictor of better quality of life [9,10]. Moreover, the practice of physical activity is one of the variables that should not be forgotten in this area, as it means an improvement in the quality of life of older people, as well as a guarantee of good ageing in terms of health [11], and should be taken into account as a measure to protect the health and functional skills that result in a better quality of life for older adults and lead to better physical health and improved functional capacity [10].

Active and healthy aging is related to the development and improvement of the functional capacity of the population, and interventions such as those based on reminiscence therapy can be considered. Interventions such as those that promote proximity and memories of the past, based on principles of reminiscence, along with a focus on interdisciplinarity, can be a contribution for older people regarding quality of life, health, social relations, and staying active [12]. A systematic review of the literature concluded that reminiscence-based interventions have several benefits in older adults, such as improvements in cognitive function, anxiety, depressive symptoms, self-esteem, life satisfaction, and personal interaction [13].

It is also important to highlight that person-centered care is increasingly recommended in mental health care, and the person’s preferences should be valued, with shared decision-making [14].

So, it was based on person-centered care, involving the person in the choice of themes for the sessions and the basic principles of reminiscence therapy that we developed “Geriatric Proximity” intervention, which is represented in Figure 1.

This intervention puts the person at the center, characterizing him/her by intrinsic capacity, considering his/her family. It aims to enhance the interaction between the institutional and community environment, creating new relationships within the institution and maintaining and strengthening the pre-existing social support network. It aims to maintain or strengthen affections, social support, and functioning, as well as to prevent feelings of loneliness. In this way, it contributes to the promotion of active and healthy aging.

To contribute to a proximity model with institutionalized older adults in a pandemic context, we intend to implement the intervention “Geriatric Proximity” with this study, knowing that it is a fuse for the countless strategies that need to be developed, where nurses should play a central role. The objective of this study is to assess the impact of the implementation of a “Geriatric Proximity” intervention on the functioning, satisfaction with social support, affective experience, and feelings of loneliness of institutionalized older adults.

## 2. Materials and Methods

This is a pilot, quantitative, quasi-experimental study.

The sample was composed of 34 institutionalized older adults from two residential homes. After acceptance of participation in the study by the heads of two residential homes in Portugal, a number was assigned to each of the institutions, and a randomization was made through a paper draw to define which intervention would be the CG and EG. The older adults from the experimental group (EG) lived in institution A (*n* = 17), and the older adults from the control group (CG) lived in institution B (*n* = 17). The EG was subjected to the intervention, and the CG was not subjected to the intervention maintaining usual care.

Both institutions are urban residential homes for the older adults (over 65 years). These types of institutions were created in Portugal for older adults, with autonomy or in a situation of loss of independence/autonomy, under the intervention of multidisciplinary technical teams, which provide biopsychosocial support and health care. Services such as food and nutrition, personal hygiene and comfort care, laundry, and nursing are provided in both institutions. Totals of seventy-one to sixty-nine people live in both residential houses. The staff in this type of institution includes administrative personnel, social workers, kitchen staff, socio-cultural animators, assistants, nurses, and a doctor who provides support when necessary.

### 2.1. Instruments

A sociodemographic and clinical questionnaire was applied to collect the following data: age, gender, marital status, level of education, and main/major areas of medical diagnosis. Four assessment instruments were used in data collection, which we describe be-low, all validated for the Portuguese population.

#### 2.1.1. Social Support Satisfaction Scale

The social support satisfaction scale (ESSS) is a scale that was developed and validated by Ribeiro (1999), with the purpose of measuring satisfaction with social support. It is composed of 15 items, which are statements in which each item should be marked on a Likert scale, with one of five ratings: “totally agree”, “mostly agree”, “neither agree nor disagree”, “mostly disagree”, and “totally disagree” [15]. These items are rated from 1 to 5, with 1 being assigned to A and 5 being assigned to E. The ESSS consists of 4 domains, being satisfaction with friends; intimacy; satisfaction with family; and social activities. However, this scale has inverted items, namely 4, 5, 9, 10, 11, 12, 13, 14, and 15, where 1 is assigned to E and 5 is assigned to A. The final score of the scale ranges from 15 to 75, and the higher the score, the better the satisfaction with social support.

#### 2.1.2. Elderly Nursing Core Set

The elderly nursing core set (ENCS) was developed by Fonseca and collaborators and aims to assess the functioning of older adults (Fonseca et al., 2019). This instrument consists of 25 questions based on the international classification of functioning, disability, and health (ICF), scored on a Likert scale from 1 to 5. These scores provide a functional profile as: 1. no disability: 0–4%; 2. mild disability: 5–24%; 3. moderate disability: 25–49%; 4. severe disability: 50–95%; 5. total disability: 96–100%. The ENCS consists of four domains that are subdivided into various ICF codes: self-care, learning and mental functions, communication, and social relationships. The higher the score, the worse the individual’s functional profile.

#### 2.1.3. Positive Affect Negative Affect Scale—Reduced Version

The third scale, the positive affect negative affect scale—reduced version—or PANAS-VRP, aims to assess positive affect (PA) and negative affect (NA), defined as general dimensions that describe the affective experience of individuals. The PANAS, original version, was developed by Watson, Clark, and Tellegen (1988), later adapted and validated for the Portuguese population by Galinha and Ribeiro (2005). It is composed of 10 items, which were answered according to a 5-point Likert scale (1. very little or not at all; 2. a little; 3. moderately; 4. quite a lot; 5. extremely). Of these 10 items, 5 refer to negative affections and 5 refer to positive affections.

#### 2.1.4. UCLA-Loneliness Scale

The last and fourth scale, Loneliness Scale, UCLA. This scale is used to find out the level of loneliness in the population, not directly asking about feelings of loneliness. The original scale, the UCLA-Loneliness Scale, was created by Russel, Repleu, and Fergunson (1978) and later adapted and validated for the Portuguese population by Pocinho and Farate. It is composed of 18 items, being a Likert-type scale, with 4 points, ranging from “Never”—1, “Rarely”—2, “Sometimes”—3, to “Very often”—4. The range of scores is between 18 and 72, and the higher the score, the higher the level of loneliness.

### 2.2. Intervention

The intervention itself followed the principles of reminiscence therapy, which, according to the creator, consists of a universal and natural mental process and is now considered a normal feature, not only in the final chapter of life. Reminiscence, as a process inherent to all human beings, consists of the recollection of an experience or fact to which the subject habitually associates pain or pleasure. It may or may not be exclusive to a specific stage of life, and it is known that it becomes progressively more present as the person gets older, allowing an individual, or in an interactional way, us to analyze the past, understand changes, adapt to transitions, acquire knowledge, communicate with others, and promote self-image [16].

The responsibility for the sessions was always the same researcher (nurse) and the occupational therapist of the EG institution, with a total of 10 sessions in each group. In each group, in the first session, they introduced themselves to the group. The intervention was conducted in group, with a maximum duration of 45 min each session. The theme of each session was previously chosen by participants to centralize care according to their preferences. Each participant freely developed his or her own process of remembering the episode of his or her life, sharing it with the group. The themes addressed were marriage; children; festive activities: 25 April (Carnation Revolution that occur in 1974 in Portugal); occupational life; school and youth; objects: cell phone; trips; dreams; festive activities: birthdays; childhood games; toys; friends; milestone day. The respective themes of each session were always proposed to the participants in a broad way, for example “Outstanding objects”, leaving them responsible for the concrete choice of the same, for example, “Cell phone”.

The materials used to help and enrich the sessions were limited to a tablet (due to the restriction on the entry of objects in institution, due to the pandemic), which was used to record the sessions, to evaluate the three moments of evaluation, and to display illustrative images, such as in the theme that addressed the experiences in festive activities; specifically, for the April 25th revolution, an illustrative image was displayed on the tablet that helped the sessions.

### 2.3. Procedures

The baseline data collection was performed in May 2021, with the explanation of the study to the participants, the signing of the informed consent, and the first occasion of assessment of the four scales. Data collection was performed face-to-face, in the activity room of residential homes, individually, with no one else inside, safeguarding the principle of confidentiality and privacy.

Soon after this first assessment, we began the group interventions on 28 May with small subgroups. GE was divided in three: two were composed of five participants, and one group consisted of seven participants. The duration of each session was around forty-five minutes, following the guidelines of previous studies with positive results.

The second and intermediate occasion of data collection took place on 23 June, and the final assessment took place on 16 July.

### 2.4. Data Analysis

To evaluate the normality of the numerical variables, the Shapiro–Wilks test was used. To evaluate the homogeneity of variances, Leneve’s test was used. To compare the age of users by group or gender, the *t*-test was used.

For each occasion of measurement, to compare the value of each domain/scale of the 2 groups, the *t*-test was used, or the Wilcoxon Mann–Whitney test, in case of violation of the normality assumption.

In the EG, mixed ANOVA was used to compare the domain/scales values at the 3 measurements, with gender as between-subject factors and moments as within-subject factors. Under departures from normality, robust ANOVA was performed. Levene’s test was used to check the homogeneity of variance assumption of the between-subject factor. The Mauchly’s test used to assess the sphericity assumption. The Greenhouse–Geisser sphericity correction was applied to factors violating the sphericity assumption. When significant differences were found, multiple pairwise comparisons with Bonferroni correction were run.

Statistical analysis was performed with the R program, and a 5% significance level was considered in the analyses.

## 3. Results

The sample was consisted of 34 participants (EG (*n* = 17); CG (*n* = 17)). Most of the EG were men (64.7%), and age ranged from 66 to 94 years. Most of the CG were women (64.7%), and age ranged from 68 to 97 years. The mean ages of the users did not differ significantly between groups (*p* = 0.463).

Widowed marital status predominated in both groups (EG: 64.7%, CG: 58.8%). In the EG, 9 out of 17 (52.9%) attended school, against just 2 out of 17 (11.8%) in the CG. Table 1 shows the characteristics of the sample.

The sample consisted of 34 participants (EG (*n* = 17); CG (*n* = 17)). Most of the EG were men (64.7%), and age ranged from 66 to 94 years. Most of the CG were women (64.7%), and age ranged from 68 to 97 years. The mean ages of the users did not differ significantly between groups (*p* = 0.463).

There was a significant difference between the baseline in the experimental group and the control group in all domains/scales (all *p* < 0.005; Table 2). The CG shows lower values than the EG in the functional profile, loneliness, and negative affect scales and higher values in the satisfaction with general social support and positive affect scales (Table 2).

In the second measurement, no significant difference was detected between the two groups in either the satisfaction with general support or in the loneliness scale (both *p* > 0.05; Table 2). At the third measurement, a significant difference was found only in the general functional profile and in negative affect (both *p* < 0.05; Table 2).

All users in the CG maintained the value of the domains/scales on the various moments of measurement.

From the first to the third measurement, there was a significant reduction in the general functional profile, loneliness scale, and negative affect, as well as an increase in satisfaction with general support and positive affect (todos os *p* < 0.05) (Figure 2).

In the CG, we observed the maintenance of the values of the different sections of the ENCS: learning of mental functions, communication, and self-care (Figure 3).

In the EG, a significant difference was detected in the learning of mental functions over the three moments (*p* = 0.008), with the value of the first measurement being higher than the third measurement; regarding communication and self-care, no significant differences were observed (Figure 3).

## 4. Discussion

The impact of implementing the “Geriatric Proximity” intervention in predominantly female older adults with associated morbidity was evaluated.

Our results showed clear differences between the pre-intervention (first measure) and post-intervention (third measure) periods in the EG that should be analyzed.

After the implementation of the intervention and observing the evolution of the general functioning profile, an improvement in the EG was identified. However, when observing the three domains of the functional profile, improvements were only observed in the domain learning of mental functions. Although there are no identical interventional studies that have used the ENCS, we can associate this domain with cognitive functions. The use of memories can be used to stimulate cognitive functioning [17]. A study that applied reminiscence therapy in a population with similar characteristics to the present study showed no significant relationship between this type of therapy and memory, executive functions, mood, or quality of life [18]. In turn, another study, which applied an intervention using reminiscence therapy, with 26 individual sessions, showed a significant effect on global cognition, memory, and quality of life [19]. These results are in agreement with our study, since the “Learning of mental functions” domain assesses cognition and memory.

The improvements identified in the EG, in relation to social support satisfaction, specifically the improvement in satisfaction with friends and social activities, may be related to a greater interaction between the group of users during the intervention, as well as to the intervention itself, which can be considered a social activity, due to the sharing and interaction within the group. Chiang et al. (2010) reported that reminiscence therapy promotes improved socialization and feelings of accomplishment, leading to greater satisfaction with social support [19]. Studies point in the same direction, referencing that the participants’ perception of social activity supports that they are performing a group social activity with a given frequency increases [16]. Moreover, these results are consistent with the fact that sharing life stories helps build friendships and stimulate group cohesion to achieve psychological comfort in the transition they experience [17]. Other studies show that institutionalized older adults benefit from participation in programs aimed at increasing the level of social support and the frequency of meaningful social interactions. Social support is, thus, associated with improved cognitive functioning, decreased depressive symptoms, and even an increased quantitative and qualitative social support network [17]. Another study reported that sharing life stories helps build friendships and stimulates group cohesion with the promotion of psychological comfort during a transition [20].

Our data also show an improvement in terms of affective experience. In the EG, we observed a decrease in the experience of negative affect and an increase in the experience of positive affect after the implementation of the intervention. Associating depression with negative affect as the disorder of affect, our data, supported by other studies, show that these types of interventions are successful in improving depressive feelings. At first, the participants were passively sharing their life experiences, and over the course of the sessions, an internal cohesion gradually developed that allowed them to actively share their reminiscences and identify meaning in that social activity [16]. The literature points to a significant increase in depression remission with reminiscence therapy-based interventions, with significant improvement in mood [21,22].

Regarding the feeling of loneliness, we found that there was a decrease in this feeling in the EG. When an older adult moves from his/her home to an institution, such as a residential home, this person may experience a greater feeling of loneliness than the one at home or even at a day care center [23]. One of the studies that aimed to analyze the effects of reminiscence therapy on feelings of loneliness and institutionalized older adults, which used a program similar to that of the present study, with sessions 1 to 3 times a week, concluded that there was a decrease in feelings of loneliness with the intervention [16]. This type of intervention gives people the opportunity to interact, rather than remain alone. The sharing that takes place between participants stimulates friendship and understanding between them, giving them a sense of acceptance in the group. Through the learning that comes from sharing other lives, participants realize that each life is unique and interesting, even if some seem sad or frustrated. The group itself builds a sense of belonging and cohesion among them that helps them overcome the feeling of loneliness. Additionally, memories enter here as a principle of this therapeutic intervention to also help them in the validation of their own “I” [16,23].

In the CG, we did not obtain significant differences in the pre- and post-tests. This may indicate that the differences, in relation to the EG, are due to the implementation of the “Geriatric Proximity” intervention, since all interventions that make up the functioning dynamics of the institutions remained stable, as well as all elderly care (hygiene and comfort, food and hydration, and sleep).

Interventions, such as the one we implemented in this research, may offer a viable approach to improving the mental health of institutionalized older adults, given that satisfaction with social support, affectivity, cognition, and decreased feelings of loneliness contribute to the promotion of mental health.

A strong point of this work is the contribution with the “Geriatric Proximity” intervention that can be applied of institutionalized older adults. The main limitation of our study was the pandemic context in which it was carried out. This was due to the impossibility of collaborating with more members of the multi-disciplinary team and with family members, as well as to the fact that, in the middle of the study, an outbreak occurred in one of the institutions, which postponed the study. The small sample size is also a limitation of this study, as well as the significant differences found at baseline between both groups.

We highlight, as implications of this study, that interventions, such as the one we implemented in this research, may offer a viable approach to improving the mental health of institutionalized older adults, given that satisfaction with social support, affectivity, and cognition and decreased feelings of loneliness contribute to the promotion of mental health.

A future line of research from this study may focus on a follow-up to understand the durability of the beneficial effects over time.

## 5. Conclusions

“Geriatric Proximity” intervention improved the learning and mental functions dimension of older adults, a domain that is associated with cognition. However, no improvements were observed in the domains of self-care and communication. Interventions, such as the one we implemented in this research, may offer a viable approach to improving the mental health of institutionalized older adults, given that satisfaction with social support, affectivity, and cognition and decreased feelings of loneliness contribute to the promotion of mental health. However, it is necessary to go further, so that we also have functioning results, in terms of self-care. This requires not only procedural and structural changes, but also an increase in resources.

## Figures and Tables

**Figure 1 geriatrics-08-00039-f001:**
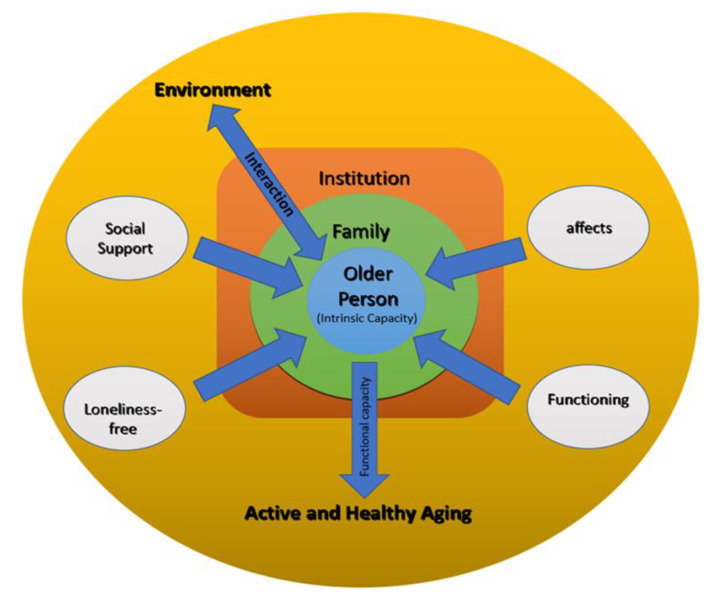
“Geriatric Proximity” intervention.

**Figure 2 geriatrics-08-00039-f002:**
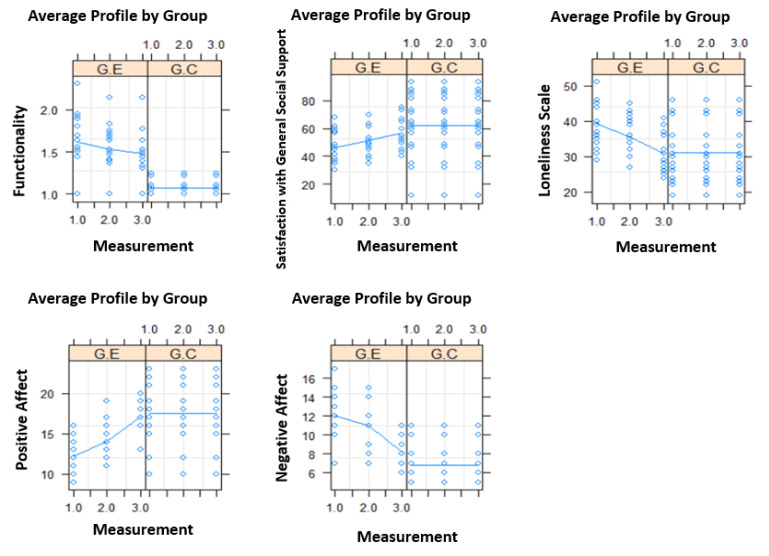
Evolution of each measure by group over the three evaluation moments and average profile of users (solid line) by group.

**Figure 3 geriatrics-08-00039-f003:**
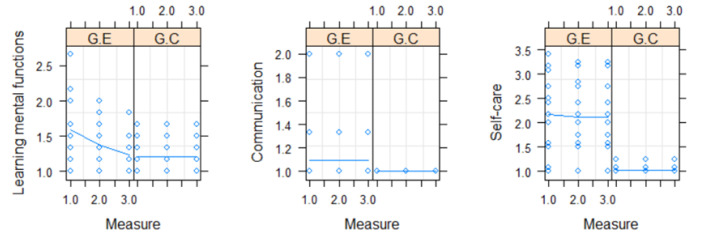
Evolution of each measure per group over the 3 evaluation moments and average user profile (solid line) per group.

**Table 1 geriatrics-08-00039-t001:** Sociodemographic Characterization.

Variables	Total Sample	EG	CG
	(*n* = 34)	(*n* = 17)	(*n* = 17)
Age	M = 81.97;	M = 80.94;	M = 83;
	SD = 8.03	SD = 8.71	SD = 7.40
Sex	34	17	17
- Female	17 (50%)	6 (35.3%)	11 (64.7%)
- Male	17 (50%)	11 (64.7%)	6 (35.3%)
Marital Status			
- Married (*n* (%))	2 (5.9%)	1 (5.9%)	1 (5.9%)
- Divorced (*n* (%))	3 (8.8%)	3 (17.6%)	0 (0%)
- Single	8 (23.5%)	2 (11.8%)	6 (35.3%)
- Widowed (*n* (%))	21 (61.8%)	11 (64.7%)	10 (58.8%)
Academic Qualifications			
- Has not attended school and cannot read or write	23 (67.6%)	8 (47.1%)	15 (88.2%)
- Attended school, but did not go to higher education	10 (29.4%)	8 (47.1%)	
- Attended higher education	1 (2.9%)	1 (5.9%)	2 (11.8%)
Multimorbidity	29 (85.3%)	13 (76.5%)	16 (94.1%)
(More than one diagnostic area)			

**Table 2 geriatrics-08-00039-t002:** Mean and standard deviation of the domains/scales before (baseline), during (2nd measure), and after (3rd measure) intervention in each group and *p*-value of the *t*-test or the Wilcoxon Mann–Whitney test (a), which compares the measurement value of the 2 groups.

Domain/Scale	Group	1st Measure	*p* Value	2nd Measure	*p* Value	3rd Measure	*p* Value
Overall functioning profile	EG	1.62 ± 0.32	<0.001 a	1.53 ± 0.28	<0.001 a	1.48 ± 0.27	<0.001 a
	CG	1.08 ± 0.07		1.08 ± 0.07		1.08 ± 0.07	
Satisfaction with social support	EG	46.47 ± 11.04	0.015	51.76 ± 10.43	0.091	56.57 ± 10.16	0.344
	CG	62.35 ± 22.45		62.35 ± 22.45		62.35 ± 22.45	
Loneliness scale	EG	39.29 ± 6.15	0.002	35.59 ± 5.50	0.057	30.94 ± 5.02	0.958
	CG	31.06 ± 7.64		31.06 ± 7.64		31.06 ± 7.64	
Positive affect	EG	12.18 ± 1.98	<0.001	14.06 ± 2.08	0.002	17.06 ± 1.92	0.668
	CG	17.47 ± 3.41		17.47 ± 3.41		17.47 ± 3.41	
Negative affect	EG	12.00 ± 2.45	<0.001 a	11.00 ± 2.47	<0.001 a	8.24 ± 1.48	0.024 a
	CG	6.82 ± 1.85		6.82 ± 1.85		6.82 ± 1.85	

a = Wilcoxon Mann–Whitney test.

## Data Availability

The data presented in this study are openly available in repository at University of Évora in that link http://hdl.handle.net/10174/31404.

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
