# Peer review of "“Geriatric Proximity” Intervention in COVID-19 Context: Contribution to Reducing Loneliness and Improving Affectivity"

_geriatrics, 2023, doi:10.3390/geriatrics8020039_

Round 1

Reviewer 1 Report (Previous Reviewer 3)

After examining the scientific study, the following considerations may be made. The scientific study is well structured in all its parts. In particular, the premises with which the authors introduced the analysis are clear. Equally clear are the objectives that led the authors to carry out this study and the section on materials and methods. Particular appreciation can also be expressed for the material on which the study was carried out. The data was collected methodically and without bias. The results were consistent and significant and allowed a discussion section full of food for thought. The authors then developed a discussion of the results achieved.

The number and quality of the citations are appropriate; however, the scientific relevance of the article could benefit from an expansion of the same. In the specific advice to add the following quotes:

·       Di Fazio N, Morena D, Delogu G, Volonnino G, Manetti F, Padovano M, Scopetti M, Frati P, Fineschi V. Mental Health Consequences of COVID-19 Pandemic Period in the European Population: An Institutional Challenge. Int J Environ Res Public Health. 2022 Jul 30;19(15):9347. doi: 10.3390/ijerph19159347. PMID: 35954706; PMCID: PMC9367746.

·       di Fazio N, Caporale M, Fazio V, Delogu G, Frati P. Italian law no. 1/2021 on the subject of vaccination against Covid-19 in people with mental disabilities within the nursing homes. Clin Ter. 2021 Sep 29;172(5):414-419. doi: 10.7417/CT.2021.2349. PMID: 34625770.

English is well structured in syntax and grammar.

Author Response

Thank you very much for your appreciation of our work. In order to improve the evidence, we cite in our work we have added the citations you suggested. Once again thank you very much for the opportunity.

Reviewer 2 Report (Previous Reviewer 2)

Dear Author, 

The article entitled: 

 "Geriatric Proximity" intervention in COVID-19 context: contri-bution to reducing loneliness and improving affectivity.

Presents in the pandemic context a study that evaluates the impact of the implementation of a geriatric proximity intervention on functioning, satisfaction with social support, affective experience and feelings of loneliness in older adults.

It is a pilot experimental and control group study. They applied instruments such as the satisfaction with social support, basic nursing set, positive and negative affect scores, as well as the UCLA loneliness scale.

The study shows that the experimental group reduced loneliness, negative affect and increased satisfaction with social support and positive affect. Thus, geriatric proximity intervention is a positive contribution towards the older person.

Introduction:

-I find no linkage in talking about the importance of family or social breakdown in pandemic context compromising health with the paragraph below on reminiscence. I don't understand what it has to do with, or find a better way to unify reminiscence. Reminiscence promotes memory and recollection of the past, but is not related to isolation, loneliness, social support and negative affect. Please see how to unify the thread of your introduction with "reminiscence".  Also look at how to thread it with "person-centred care".

Improve in the introduction that this variable is important for the quality of life of the older person, quality of life includes a multitude of psychosocial and physical variables among others. Also, I suggest that within the multidimensionality of quality of life, a small mention should be made of physical activity. In order to increase the internationalisation of your work, downloads and citations: I recommend that in the introduction you cite an important variable, incorporating the following text: "The practice of physical activity is one of the variables that should not be forgotten in this area, as it means an improvement in the quality of life of older people as well as a guarantee of good ageing in terms of health" https://doi. org/10.3390/bs12090331 "and should be taken into account as a measure to protect health and functional skills that result in a better quality of life for older people and that leads to better physical health and improved functional capacity" https://doi.org/10.3390/socsci11060265

Intervention

Specify what you did in those 10 sessions, what sessions you did.

You have done an intervention, but you do not describe it.

Indicate what the intervention is based on, how long each session is, if it was done in a group.

Discussion

-Towards the end of your discussion I suggest that you incorporate a few paragraphs in response:

(a) what theoretical implications does this work have for scientists reading this work, for theorists in the field or colleagues?

b) strengths of your work relative to other studies.

c) What practical implications does this work have for older people?

d) future line of research arising from this work that needs to be covered.

I hope you get better visibility this way! Congratulations, I loved your work!

My sincere congratulations for the work.

Author Response

Thank you very much for your appreciation of our work and for your comments. We improve the introduction with your suggestions.

To clarify the intervention, we added information according your suggestion.

We added the information that you suggest in discussion.

Round 2

Reviewer 1 Report (Previous Reviewer 3)

After examining the scientific study, the following considerations may be made. The scientific study is well structured in all its parts. In particular, the premises with which the authors introduced the analysis are clear. Equally clear are the objectives that led the authors to carry out this study and the section on materials and methods. Particular appreciation can also be expressed for the material on which the study was carried out. The data was collected methodically and without bias. The results were consistent and significant and allowed a discussion section full of food for thought. The authors then developed a discussion of the results achieved.

This manuscript is a resubmission of an earlier submission. The following is a list of the peer review reports and author responses from that submission.

Round 1

Reviewer 1 Report

Thank you for the opportunity to review this manuscript. This manuscript presents a small pilot study addressing an important topic to the wellbeing and quality of life of institutionalised older people.

While the topic is noble, I do have some concerns about the quality of the manuscript, mostly in relation to lack of information provided in the methods section, and the presentation of the result section.

In methods section, the authors need to give a better description of the intervention. It also needs to be explained why it states that the control group received no intervention, but were then grouped into three groups during the intervention period? 

The authors state that allocation to intervention and control was random, but offer no explanation how the randomisation was done.

The authors do not describe how the care homes were selected, or provide any contextual information to describe the homes (e.g. rural/urban, size of home/ number of residents/ the number of staff, specialities e.g. a dementia). 

The statistical analysis mentions ANOVA and adjusting for 'occasions'- what is occasions?

The results section is confusing to read and requires to be re-written with careful attention to the tables being used and what they should be conveying. 

Please ensure you add footnotes e.g. Table 3 has an 'a' in the table but no explanation below it. Table 3 also has two columns for 2nd measure, without explanation for this.

While this is an important and timely topic to address, there are a number of methodological points requiring to be addressed in the manuscript and the results section requires restructuring to ensure the story is clear to readers.

In general use of language is good but there were a few random word choices; suggest checking the literature for conventional terms in English used in scientific papers. 

I think the discussion is a bit long. Please try to  reduce and stick to pertinent points.

Author Response

In methods section, the authors need to give a better description of the intervention.

Thank you for your comment. We add more information about intervention.

It also needs to be explained why it states that the control group received no intervention, but were then grouped into three groups during the intervention period?

It was a mistake. Thank you for alert. We remove it. Only EG was grouped into three groups.

The authors state that allocation to intervention and control was random but offer no explanation how the randomisation was done.

We add in methods this explanation: “. After acceptance of participation in the study by the heads of two residential homes in Portugal, a number was assigned to each of the institutions and a randomization was made through a paper draw to define which intervention would be the CG and EG.”

The authors do not describe how the care homes were selected, or provide any contextual information to describe the homes (e.g. rural/urban, size of home/ number of residents/ the number of staff, specialities e.g. a dementia).

We add this informations: "Both institutions are urban residential homes for the older adults (over 65 years). These types of institutions were created in Portugal for older adults, with autonomy or in a situation of loss of independence/autonomy, under the intervention of multidisciplinary technical teams, which provide biopsychosocial support and health care. Services like Food and Nutrition, Personal Hygiene and Comfort Care, Laundry and Nursing are provided in both institutions. In both residential houses live seventy-one to sixty-nine people. The staff in this type of institution includes administrative personnel, social workers, kitchen staff, socio-cultural animators, assistants, nurses, and a doctor who provides support when necessary."

The statistical analysis mentions ANOVA and adjusting for 'occasions'- what is occasions?

It was a translation oversight.

We have eliminated the word occasions.

We sent the article for linguistic review, but in order not to delay the review process we submitted the article with the changes suggested by the reviewers and will send the article later with the revised English.

The results section is confusing to read and requires to be re-written with careful attention to the tables being used and what they should be conveying.

We change this section.

Please ensure you add footnotes e.g. Table 3 has an 'a' in the table but no explanation below it. Table 3 also has two columns for 2nd measure, without explanation for this.

We add footnotes and change 2nd measure to 3rd measure.

While this is an important and timely topic to address, there are a number of methodological points requiring to be addressed in the manuscript and the results section requires restructuring to ensure the story is clear to readers.

In general use of language is good but there were a few random word choices; suggest checking the literature for conventional terms in English used in scientific papers.

I think the discussion is a bit long. Please try to reduce and stick to pertinent points.

We have taken the suggestion for improvement into account and cut out the fourth paragraph.

Reviewer 2 Report

Dear authors, 

The article by title: 

 "Geriatric Proximity intervention in COVID-19 context: contribution to reducing loneliness and improving affectivity".

It aims to evaluate the impact of the implementation of a geriatric proximity intervention.

I will then make some suggestions for improvement, with the aim of improving its visibility, citations, downloads and internationalisation.

In the introduction:

After the third line of your introduction I suggest you incorporate these references to increase the internationalisation of your study, citations and downloads:

"being necessary to take into account among the relevant elements for older people in the quality of life, health, social relations and staying active, being physical activity one of the most important disease prevention and health promotion strategies on a physical and cognitive scale " https://www.mdpi.com/2254-9625/7/3/135  

"Those people over 65 years of age who practice a high level of physical activity are the ones who consume more leisure time and increase social relations, increasing social activities as an element of participation and quality of life, being the practice of physical activity one of the ways to increase social relations and social activities." https://revistes.ub.edu/index.php/Anuario-psicologia/article/view/anpsic2020.50.12

Older people who are physically active have a good perception of their health.The gender variable is therefore a highly relevant variable to be studied https://doi.org/10.20318/recs.2017.4002

Towards the end of your discussion I suggest you incorporate a few paragraphs in response:

(a) what theoretical implications does this work have for scientists reading this work, for theorists in the field or colleagues?

b) strengths of your work in relation to other studies.

c) What practical implications does this work have for older people?

d) future line of research arising from this work that needs to be covered.

I hope you get better visibility this way! Congratulations, I loved your work!

My sincere congratulations for the work.

Author Response

In the introduction:

After the third line of your introduction, I suggest you incorporate these references to increase the internationalisation of your study, citations and downloads:

"being necessary to take into account among the relevant elements for older people in the quality of life, health, social relations and staying active, being physical activity one of the most important disease prevention and health promotion strategies on a physical and cognitive scale " https://www.mdpi.com/2254-9625/7/3/135 

We have taken the suggestion for improvement into account and we add this reference.

"Those people over 65 years of age who practice a high level of physical activity are the ones who consume more leisure time and increase social relations, increasing social activities as an element of participation and quality of life, being the practice of physical activity one of the ways to increase social relations and social activities." https://revistes.ub.edu/index.php/Anuario-psicologia/article/view/anpsic2020.50.12

(Parra-rizo, 2020)

Parra-rizo, M. A. (2017). Diferencias de género en la percepción de salud en personas mayores de 60 años físicamente activas Gender differences in the perception of health in physically active people over 60 years of age. 8(2), 219–227.

Parra-rizo, M. A. (2020). Anuario de The UB Journal of Psychology | 50/3. https://doi.org/10.1344/anpsic2020.50.12

We have taken the suggestion for improvement into account and add this reference.

Older people who are physically active have a good perception of their health. The gender variable is therefore a highly relevant variable to be studied https://doi.org/10.20318/recs.2017.4002

(Parra-rizo, 2017)

Parra-rizo, M. A. (2017). Diferencias de género en la percepción de salud en personas mayores de 60 años físicamente activas Gender differences in the perception of health in physically active people over 60 years of age. 8(2), 219–227.

Parra-rizo, M. A. (2020). Anuario de The UB Journal of Psychology | 50/3. https://doi.org/10.1344/anpsic2020.50.12

We have taken the suggestion for improvement into account and add this reference.

Towards the end of your discussion, I suggest you incorporate a few paragraphs in response:

(a) what theoretical implications does this work have for scientists reading this work, for theorists in the field or colleagues?

“Geriatric Proximity" intervention improved the learning and mental functions dimension of older adults, a domain that is associated with cognition. However, no improvements were observed in the domains of self-care and communication. We believe that, for there to be an effective improvement in these areas, it is not enough to apply an intervention program, but a set of changes in the process and structure of the institutions is necessary, through the implementation of an integrated care model. This model needs a reinforcement of specialized human resources, such are the current demands in terms of health care of the institutionalized older adults.

b) strengths of your work in relation to other studies.

We have "Interventions, such as the one we implemented in this research, may offer a viable approach to improving the mental health of institutionalized older adults, given that satisfaction with social support, affectivity, cognition, and decreased feelings of loneliness contribute to the promotion of mental health. However, it is necessary to go further so that we also have functioning results in terms of self-care. This requires not only procedural and structural changes, but also an increase in resources."

c) What practical implications does this work have for older people?

We have: "A geriatric outreach intervention, based on principles of reminiscence therapy, thus proves to be an intervention with the potential to maintain or even increase the psychosocial health and well-being of the older adults."

d) future line of research arising from this work that needs to be covered

We added in the conclusion: "A future line of research resulting from this work that needs to be covered is to conduct a follow up study to understand the time duration of the effects of the application of the intervention as to understand if it continues to be applied. "

Reviewer 3 Report

After examining the scientific study, the following considerations may be made. The scientific research is well structured in all its parts. In particular, the premises with which the authors introduced the analysis are clear. The objectives that led the authors to carry out this study and the section on materials and methods are equally clear. Particular appreciation can also be expressed for the material on which the study was carried out. The data was collected methodically and without bias. The results were consistent and significant and allowed a discussion section full of food for thought. The authors then developed a discussion of the results achieved. 

 This article is particularly current as it is particularly relevant to the issue of the covid-19 pandemic.

The number and quality of the citations are appropriate. However, the scientific relevance of the article could benefit from expansion of the same. In the specific advice to add the following quotes:

·       Introduction - Line 9. I suggest adding the following sentence “ Other European countries introduced laws for the Sars-Cov-2 vaccination of old people in nursing homes” as well as the following quote: “di Fazio, N., Caporale, M., Fazio, V., Delogu, G., & Frati, P. (2021). Italian law no. 1/2021 on the subject of vaccination against Covid-19 in people with mental disabilities within the nursing homes. La Clinica terapeutica172(5), 414–419. https://doi.org/10.7417/CT.2021.2349

English is well structured in syntax and grammar.

Author Response

After examining the scientific study, the following considerations may be made. The scientific research is well structured in all its parts. In particular, the premises with which the authors introduced the analysis are clear. The objectives that led the authors to carry out this study and the section on materials and methods are equally clear. Particular appreciation can also be expressed for the material on which the study was carried out. The data was collected methodically and without bias. The results were consistent and significant and allowed a discussion section full of food for thought. The authors then developed a discussion of the results achieved.
This article is particularly current as it is particularly relevant to the issue of the covid-19 pandemic.
The number and quality of the citations are appropriate. However, the scientific relevance of the article could benefit from expansion of the same. In the specific advice to add the following quotes:
Introduction - Line 9. I suggest adding the following sentence “Other European countries introduced laws for the Sars-Cov-2 vaccination of old people in nursing homes” as well as the following quote: “di Fazio, N., Caporale, M., Fazio, V., Delogu, G., & Frati, P. (2021). Italian law no. 1/2021 on the subject of vaccination against Covid-19 in people with mental disabilities within the nursing homes. La Clinica terapeutica, 172(5), 414–419. https://doi.org/10.7417/CT.2021.2349”

We appreciate the evaluation of our article and to enrich our work we have added the suggested bibliography. 

Round 2

Reviewer 1 Report

The authors have kindly taken steps to improve the use of language and grammar in the text, and added to their methods section as I had suggested. It is still unclear how the 2 nursing homes were selected in the first place, but all other methods queries have been dealt with.

The results section for me still lacks clarity. Given the significant difference at baseline between the two groups, it is not clear to me why the authors continue to report between group measures here. Information on trends in each group, and the difference in trend between the two groups, would be meaningful. As it stands, the gradual improvement in the experimental group could just be regression to the mean.

I do not recall seeing figures 1-3 in the original submission. Apologies if I missed it.

The discussion and conclusion are still very long and poorly structured.

The addition of references in relation to physical activity and vaccination laws feel random and disconnected from this paper. For me, as indicated my previous comments to the editor, I am a bit suspicious of self-citation from a peer reviewer. I see no improvement to the paper through the inclusion of these references.

My impression from second round is that this manuscript should be rejected on the basis of its low quality.

Reviewer 3 Report

After examining the scientific study, the following considerations may be made. The scientific research is well structured in all its parts. In particular, the premises with which the authors introduced the analysis are clear. The objectives that led the authors to carry out this study and the section on materials and methods are equally clear. Particular appreciation can also be expressed for the material on which the study was carried out. The data was collected methodically and without bias. The results were consistent and significant and allowed a discussion section full of food for thought. The authors then developed a discussion of the results achieved.
This article is particularly current as it is particularly relevant to the issue of the covid-19 pandemic.